# Plasmonic Nanocomposite Implants for Interstitial Thermotherapy: Experimental and Computational Analysis

**DOI:** 10.3390/ma14040841

**Published:** 2021-02-10

**Authors:** Yvonne Kafui Konku-Asase, Kwabena Kan-Dapaah

**Affiliations:** Department of Biomedical Engineering, School of Engineering Sciences, University of Ghana, Legon Accra P.O. Box LG 77, Ghana; yvonnekafui@gmail.com

**Keywords:** plasmonic nanocomposite implants, interstitial thermotherapy, photothermal heating, breast cancer, finite element method

## Abstract

The ferromagnetic implant (thermoseeds) technique offers desirable features for interstitial thermotherapy. However, its efficacy has been reported to be limited by issues that are related to the properties of the metal alloys that are used to fabricate them and the high number of thermoseeds needed to achieve therapeutic temperature levels. Here, we present the results of a combination of experimental and computational analysis of plasmonic nanocomposite implants (photoseeds)—a combination of Au nanoparticles (NPs) and poly-dimethylsiloxane (PDMS)—as a model material. We performed structural and optical characterization of the Au NPs and repared Au-PDMS nanocomposites, followed by an elucidation of the heat generation capabilities of the Au-PDMS photoseeds in aqueous solution and in-vitro cancer cell suspension. Based on the experimental results, we developed a three-dimensional (3D) finite element method (FEM) model to predict in-vivo thermal damage profiles in breast tissue. The optical absorbance of the Au-PDMS photoseeds were increasing with the concentration of Au NPs. The photothermal measurements and the in-vivo predictions showed that the photothermal properties of the photoseeds, characteristics of the laser sources, and the duration of heating can be tuned to achieve therapeutic temperature levels under in-vitro and in-vivo conditions. Collectively, the results demonstrate the feasibility of using photoseeds for interstitial thermotherapy.

## 1. Introduction

Interstitial thermotherapy (IT) involves the elevation of tumor temperature to hyperthermic (41–45 ${}^{\circ }$C) or ablative (>50 ${}^{\circ }$C) levels while using implantable devices. The ferromagnetic seeds (thermoseeds) technique is an IT modality that has been described in the literature as safe and effective [1] due to characteristics, such as the limited collateral damage to surrounding healthy tissue due to the localization of treatment, short heating time, and uniform heat distribution [2]. First, it involves the implantation of an array of the thermoseeds—small rods with diameter and length of 1 mm and a millimeter or a few centimeters, respectively [3]—into the tissue using either open surgery or percutaneous insertion under the guidance of several imaging techniques, such as ultrasonography, fluoroscopy, magnetic resonance imaging, and computed tomography [4]. This is followed by exposure to an extracorporeal AC magnetic field that causes the thermoseeds to generate heat that is transferred to the tumor by conduction [4,5]. Their temperature self-regulating mechanism, which ensures that the temperature does not exceed a critical point—Curie transition temperature (CTT)—is a desirable property for thermotherapy. The feasibility and effectiveness of the technique have been demonstrated in single or multimodal studies that range from fundamental research through to clinical [2,6].

Conventionally, thermoseeds have been fabricated with metal alloys that are based on Fe, Cu, Ni, and Co. Although alloys made from these metals have been made to have CTTs in the therapeutic range (40–100 ${}^{\circ }$C) [3,4,5], a major drawback that affects the efficacy of the treatment and precludes their clinical use is related to the composition of the alloys. For example, alloys of Fe and Cu are not biocompatible, whereas Ni-based alloys have been reported to exhibit poor corrosion resistance [7], potentially cause hypersensitivity [8], and exhibit inconsistent CTTs [9]. To address these issues, alloys with different compositions have been examined while using in-vitro and in-vivo (animal and human) studies. Corrosion analysis of NiCu and PdCo alloys using in-vitro experiments by Paulus et al. [7] concluded that PdCo—compared to NiCu—alloys are more suited for use as a long term implants due to their excellent corrosion resistance. Furthermore, PdCo thermoseeds have been shown to provide suitable features [10,11], such as sufficient power output to treat tumors with high blood flow, consistent CTT, and reduced risk of whole-body heating due to the *Hf* factor, which does not exceed a limit—experimentally estimated to be $5\times 10^{9}$ A m${}^{-1}$s${}^{-1}$ [12]. Vallet-Regí’s [13] studied glass–glass thermoseeds synthesized from mixtures of a melt-derived glass with composition ratio: 40:40:20 mol.% of SiO${}_{2}$:CaO:Fe${}_{2}$O${}_{3}$ and a sol–gel glass with composition SiO${}_{2}$(58)–P${}_{2}$O${}_{5}$(6)–CaO(36) (mol.%) and concluded that, although the thermoseeds exhibited promising bioactive properties, the presence of glass ceramic grains on the surface hindered the bioactivity process.

Nanotechnology-assisted photothermal therapy (N-PTT) is a minimally invasive cancer treatment modalitiy that can potentially address challenges that are associated with the inductively heated ferromagnetic implant technique [14,15]. It has been receiving a lot of attention recently due to its potential to reach deep tissue coupled with a reduced effect of non-selective cell death on the healthy tissue surrounding it [15,16]. Furthermore, recent technological advances in laser delivery equipment offer the opportunity for the development of more efficient, safe, and affordable interstitial strategies [14,15,16]. N-PTT relies on the ability of nanoparticles (NPs) that are embedded within the target tissue to absorb and convert extracorporeal near infrared (NIR) light to heat for therapeutic purposes [14,15,16] Traditionally, plasmonic NPs, such as Au, Ag, Cu, as well as carbon nanotubes or graphene, have been exploited for N-PPT, due to the opportunity to manipulate their localized surface plasmon resonance (LSPR) to enhance their absorption capabilities [16,17]. This confines heating to the location of the plasmonic nanomaterials and reduces, but does not eliminate, heating of the surrounding tissue. Recently, several groups have reported the use of ceramic (iron-oxides) nanostructures, such as Fe${}_{3}$O${}_{4}$ and γ-Fe${}_{3}$O${}_{3}$, for PTT. Chu et al. [18] showed that various shapes of Fe${}_{3}$O${}_{4}$ NPs were able to kill cancer cells and tumors in in-vitro and in-vivo models, respectively. In another study, Espinosa and co-workers [19] demonstrated the ability of Fe${}_{3}$O${}_{4}$ NPs to act simultaneously as magnetic and photothermal agents, so-called photothermal agents.

Nanocomposites, which are materials that are formed by incorporating nanoparticles as fillers into polymer-based matrices were first reported in the literature in 1961 by Blumstein [20] in order to improve the thermal stability of PMMA. In the last few decades, their use as stimuli-responsive multifunctional materials has played a major role in opening up new frontiers in several fields in biomedical engineering such as drug delivery, tissue engineering, thermotherapy and biosensing [21,22,23] For example, functionalized hydrogel nanocomposites have been exploited for several targeted drug delivery applications for cancer treatment [22]. Furthermore, several investigators have reported the use of polymethylmethacrylate (PMMA)-based nanocomposites as bone cements for several application in orhtopaedics and tissue engineering applications [24]. Our group recently reported a novel nanocomposite probe for killing of cancer cells [23]. The probe is essentially a cannula with a distal heat generating magnetic nanocomposite tip and a proximal insulated shaft. Bonyár and co-workers [25] demonstrated the use Au/Ag Poly-dimethylsiloxane (PDMS) films to enhance the sensitivity of substrates that are used for surface enhanced Raman spectroscopy based biosensing applications.

Au NPs have been the prime candidates for photothermal therapy, due characteristics, such as their tunable optical properties and biocompatibility [26]. Several groups have reported fundamental research studies using in-vitro and in-vivo experiments. Recently, Rastinehad et al. [27] reported a pilot clinical trial using Au nanoshell for localized photothermal ablation of prostate tumor. PDMS is a silicone-based elastomer that possesses an attractive combination of properties, including inertness, biocompatibility, optical transparency, and high elasticity. Because of these material properties and its ease of fabrication, it has been widely utilized to prepare different nanocomposites for several biomedical applications [28]. In this study, we explored the feasibility of Au-PDMS nanocomposite implants (photoseeds) for IT. First, we studied the optical and structural properties of the Au NPs and Au-PDMS. Second, the heating performance of the fabricated cubic photoseeds was investigated in aqueous solutions and in-vitro cancer cell suspensions. Finally, the in-vivo performance of the cylindrical photoseeds was studied by simulating the heating of breast tissue with a three-dimensional finite element method (3D FEM) model, which was validated with experimental temperature measurements in aqueous solution.

## 2. Methods

### 2.1. Experimental Analysis

Materials Characterization. The morphologies of the Au NPs (99.97+%, 28 nm, US Research Nanomaterials Inc., Houston, TX, USA) were studied with a transmission electron microscope (TEM, Philips CM10, Philips Electron Optics, Eindhoven, The Netherlands). The absorption spectra of molten Au-PDMS nanocomposites were acquired with a Xenon lamp-based UV-vis-NIR spectrophotometer (Evolution 300, Thermo Fisher Scientific, Waltham, MA, USA). A Fourier transform infrared spectroscopy (FTIR, Tensor 27, Bruker Inc, Madison, WI, USA) was used to investigate the functional groups of the nanocomposite samples. The scan wave number was in the range of 600 to 3000 cm${}^{-1}$.

Nanocomposite preparation and Implant Fabrication.We prepared the Au-PDMS nanocomposites by varying the weight fraction of the Au NPs ($\phi_{\mathrm{Au}}$) within pristine PDMS (Sylgard 184 silicone elastomer kit, Dow Corning Corp., Auburn, MI, USA) matrix using the conventional blend mixing method according to the manufacturer’s specifications, as follows: we (a) mixed the base and curing agent in a 10:1 ratio by weight, (b) stirred with spatula for about 20 min. to produce a uniform mixture with adequate cross-linking, (c) degassed for 15 min in a glass beaker that was connected to a vacuum pump to remove bubbles, (d) added NPs to the mixture, stirred for another 10 min, and then degassed again for 15 min, (e) poured mixture into cubic acrylic moulds with a side length of 0.5 cm, and (f) cured at 100 ${}^{\circ }$C in an oven for 35 min. The two types of nanocomposites that were fabricated in the study were designated as AuNP-5 and AuNP-10, according to the $\phi_{\mathrm{au}}$ of 5%, and 10%, respectively. AuNP-0, pristine PDMS, was studied as a control.

Photothermal Measurements. Figure 1 shows the experimental setup used for photothermal measurements. The sample (photoseed in 0.5 mL of deionized water or suspension of cells) contained in a 1.5 mL Eppendorf tube was irradiated by an NIR continuous laser at a wavelength, λ, of 810 nm (Photon Soft Tissue Diode Laser, Zolar Technology & MFG, Canada) with an external adjustable power, $P_{0}$ (0–3 W). The distance between the sample and laser was 4–5 cm and the laser spot diameter was 1 mm. The laser power range was between 0.5–1.0 W. Each sample was identically exposed for 5 min. The surface temperatures were also continuously captured with an infrared camera (E5 Thermal Imager, FLIR Systems Inc., Wilsonville, OR, USA). Additionally, the measurements were obtained in triplicates, except stated otherwise.

In order to evaluate the heat generation capabilities, the photothermal conversion efficiency, η was calculated based on the formulation that was developed by Roper et al. [29], which is expressed as
(1)$$\eta =\frac{Q}{P_{0}\left(1-10^{-A_{810}}\right)}=\frac{m_{w}c_{p}}{P_{0}\left(1-10^{-A_{810}}\right)}\frac{\Delta T}{\Delta t}$$
where *Q* represents the heat that is dissipated by electron-phonon relaxation of plasmons on the Au surface induced by the laser irradiation of Au NPs at λ=810 nm assuming that heat dissipated from light absorbed by the Eppendorf tube, conduction, and radiation were negligible. $m_{w}$ (g) and ($c_{p}=4.185$ J g${}^{-1}$
${}^{\circ }$C${}^{-1}$) are the mass and constant pressure specific heat capacity of water, respectively. *T* (${}^{\circ }$C) represents temperature and *t* (s) time. $A_{810}$ is the absorbance at λ=810 nm.

Cell Culture and Viability. 20 μL of $1\times 10^{5}$ MDA-MB-231 cells (American Type Culture Collection, Manassas, VA, USA) were cultured each in 75 cm${}^{2}$ CELLTREAT tissue culture flask (T75 flask), (Pepperell, MA, USA) at 37 ${}^{\circ }$C. This was done under normal atmospheric pressure levels in a Leibovitz’s L-15 medium (Thermo Fisher Scientific, Inc., Waltham, MA, USA), which was supplemented with 100 I.U. mL${}^{-1}$ penicillin/100 μg mL${}^{-1}$ streptomycin and 10% FBS to form a “L15+" medium (Sigma-Aldrich, St. Louis, MO, USA). The cells were incubated for 72 h to obtain about 70% confluent cells in the tissue culture flask. Subsequently, the cell samples were rinsed with sterile DPBS (Fisher Scientific, Pittsburg, PA, USA), followed by Trypsin-EDTA solution (Fisher Scientific, Pittsburg, PA, USA) in order to reduce the concentration of divalent cations and proteins that inhibit trypsin action. The solutions in the flask were kept in the incubator for 2 min. This was done to detach the cells from the flask surface. L15+ medium was added and the combination of cells in these solutions were centrifuged to count and determine the number of live cells present. The resulting pellet from the centrifugation was resuspended in 1 mL of the L15+ medium. For photothermally heated cells, their viability was evaluated with the Trypan Blue Exclusion (TBE) assay that is described in the next section.

Trypan Blue Exclusion (TBE) Assay. Cytotoxicity was measured using the TBE assay method. Trypan blue is a molecule that is able to enter cells with damage to their membranes. Trypan blue stock solution was added to cell suspensions. After that, the resulting solutions were loaded in a hemacytometer and then examined under an optical microscope at low magnification. Cell viability (CV) was determined with the following expression
(2)%CV=[1.0−(viablecell÷totalcells)]×100
where viable cells are cells that were taken up by the trypan blue dye, which causes them to turn blue. Total cells, which represents the total number of cells counted before application of the laser. The relative CV was measured by comparison with a control, which corresponds to a condition without a photoseed (cells only).

### 2.2. Computational Analysis

The COMSOL Multiphysics 4.3a Software package (Comsol, Burlington, MA, USA) was used in order to develop a finite element method (FEM) model of biological tissue that were embedded with photoseeds and irradiated with a continuous wave Gaussian NIR laser beam. The model, which combined optical and thermal effects, was then used to predict photothermal profiles and cell death.

Geometry. The geometry that was used in the FEM model, as shown schematically in Figure 2, consisted of tissue embedded with cylindrical photoseeds. According to Ref. [30], such models give a better illustration of the main thermal phenomena. The tissue was configured as a multi-layered block of tissue with proportions being assigned according to the Breast Imaging Reporting and Data System (BIRADS) developed by American Cancer Research [31]. 4 cylindrical photoseeds were centrally arranged in a 2×2 array within the tissue block. The space (*s*) between the photoseed array was 0.5 cm. The diameter ($d_{s}$) and length (*L*) of the photoseed were 1 mm and 1.5 cm, respectively. P1 is the control point where temperatures were recorded. The assigned optical, thermal, and physical properties of different tissue layers were approximate values that were obtained from the literature. The material properties of the photoseeds were assumed to represent AuNP-10. Its absorption coefficient, $\mu_{a}$, was calculated from the UV-vis-NIR absorbance at λ=810 nm ($A_{810}$) while using the expression [32]:(3)$$\mu_{a}=2.303A_{810}/l$$
where l=30μm is the thickness of the sample.

Light Distribution. The optical diffusion approximation of the transport theory [33] was used to describe light distribution due to the dominance of scattering over absorption in biological tissues. Assuming that the light source was a continuous wave Gaussian NIR laser beam that was incident onto the breast model, the φ can be defined by
(4)$$\phi \left(\vec{r}\right)=\frac{P_{0}\exp\left(-\mu_{\mathrm{eff}}\vec{r}\cdot \hat{n}\right)}{4\pi Dr}$$
where $P_{0}$ is the laser power and $\hat{n}$ is the direction of the beam. Table 1 presents a summary of the values of the optical properties of the tissues, which were obtained from Ref. [34].

Temperature Distribution. The Pennes bio-heat transfer equation [35] was used to estimate the temperature distribution. The resulting equation is given by:(5)$$\rho c_{p}\frac{\partial T}{\partial t}=\lambda \left(T\right)\nabla^{2}T+\rho_{b}c_{b}\omega_{b}\left(\Omega \right)\left(T_{b}-T\right)+Q_{\mathrm{met}}+Q$$
where ρ (kg m${}^{-3}$) is the density, $c_{p}$ (J kg${}^{-1}$ K${}^{-1}$) is the specific heat capacity at constant pressure. λ(T) (W m${}^{-1}$ K${}^{-1}$) is the temperature dependent thermal conductivity, which is assumed to be a linear function that is defined by [36]:(6)$$\lambda \left(T\right)=\lambda_{\left(37{\;}^{\circ }C\right)}\left[1+0.0028\left(T-293.15K\right)\right]$$
where *T* (K) and $T_{b}$ (K) are the normal body and arbitrary temperatures, respectively. $\rho_{b}$ is the density of blood, $c_{b}$, the specific heat capacity of blood, and $\omega_{b}\left(\Omega \right)$ is the coefficient of blood perfusion that is assumed to be dependent on the cell damage, Ω, and is defined by [37,38]:(7)$$\begin{array}{cc} \omega_{b}\left(\Omega \right) & =\left\{\begin{array}{cc} \omega_{b}^{0} & \mathrm{if}\;\Omega =0\\ \left(1+25\Omega -260\Omega^{2}\right)\omega_{b}^{0}, & \mathrm{if}\;0<\Omega \leq 0.1\\ \left(1-\Omega \right)\omega_{b}^{0}, & \mathrm{if}\;0.1<\Omega \leq 1\\ 0, & \mathrm{if}\;\Omega >1 \end{array}\right. \end{array}$$$\omega_{b}^{0}$ ($s^{-1}$) is the baseline coefficient of blood perfusion. $Q_{\mathrm{met}}$ (W m${}^{-3}$) is the metabolic heat. *Q* accounts for external heat sources, which varies for the different domains of geometry. The heat that is generated after the absorption of NIR light is defined as $\mu_{a}\varphi \left(r\right)$ (W m${}^{-3}$) and $N\sigma_{a}\varphi \left(r\right)$ (W m${}^{-3}$) for all domains. Table 2 presents a summary of the values of the thermo-physical properties that were used in the simulation were obtained from the literature [38,39].

Thermal Damage. The Arrhenius injury model was used to estimate tissue destruction. The model, which relates temporal temperature to cell death, is defined by [40]:(8)$$\Omega \left(\tau \right)=A\int_{0}^{\tau }\exp\left(\frac{-E_{a}}{RT(t)}\right)dt$$
where $E_{a}$ (J mol${}^{-1}$) is the activation energy, *A* (s${}^{-1}$) is a scaling factor, and R=8.3 (J mol${}^{-1}$K${}^{-1}$) is the gas constant. The values for $E_{a}$ and *A* were obtained from Ref. [38] as 302 kJ mol${}^{-1}$ and 1.18 × 10${}^{44}$ s${}^{-1}$, respectively. Ω = 1 corresponds to the 100% irreversible cell damage.

Model Validation. Finally, experimental measurements were made using a water model in order to validate the numerical model. A single cubic AuNP-10 photoseed (volume = 0.125 cm${}^{3}$) was inserted into 0.5 mL distilled water in a 1.5 mL Eppendorf tube. The samples were allowed to equilibrate at room temperature (≈25 ${}^{\circ }$C) prior to their exposure to irradiation by a 1 W (810 nm) continuous near-infrared laser for 5 min. Three water samples were heated with the same photoseed. Temperature measurement was made at the bottom surface of the photoseed. The computational model calculations were made following the same experimental protocol. The baseline temperature for the calculations was 25 ${}^{\circ }$C, instead of the 37 ${}^{\circ }$C used in the main simulations. The calculated temperatures were directly compared with the experimental measurements.

Implementation. All of the properties and dimensions were explicitly added to the FEM model as parameters and variable under the “Global Definition” and “Model” nodes, respectively. The geometry was drawn with the “ellipsoid”, “block”, “sphere”, and “cylindrical” shapes, as well as the “difference” and “union” transform functions.

The temperature distribution was achieved using the bio-heat heat transfer application mode. Additionally, 37 ${}^{\circ }$C was taken as the initial temperature of the model, and all of the boundary conditions were specified as those outlined above. The heat source was added to the bio-heat transfer application mode as a user-defined heat source.

We resolved the model with successively smaller element sizes and compared the results, until an asymptotic behavior of the solution emerged, in order to enhance the accuracy of results. The comparison was done by analyzing the temperature at the interface between photoseed and the tissue. The mesh size for all calculations was defined as a physics-controlled mesh with the element size specified as “extra fine” (element sizes: maximum = 0.42 cm and minimum = 0.018 cm) and “extremely fine” (element sizes: maximum = 0.24 cm and minimum = 0.0024 cm) for the tissue and photoseed domains, respectively. This resulted in 91,557 tetrahedral; 4952 triangular; 236 edge; and, 11 vertex elements. The degrees of freedom were 126,430. The numerical solutions were obtained using the time-dependent solver “GMRES” with its default settings. The simulations were run on a mid-range workstation with Intel(R) Xeon(R) E5-1620 CPU and 8GB of RAM.

The numerical solution was broken down into three steps: (i) The volumetric power output was calculated using Equation (Equation 4), (ii) the temperature distribution was determined as a function of time, using the output from step (i) as heat generation term in Equation (Equation 5), and (iii) the thermal dose was calculated as a function of time, using the temperature history, and it was used as the input to Equation (Equation 8). Time-dependent studies were carried out in all of the finite element analyses.

## 3. Results

### 3.1. Experimental Analysis

Materials Characterization. TEM revealed the spherical morphology of the AuNPs (Figure 3a) and NP size distribution with an average diameter of 28 nm. UV-vis-NIR spectra (Figure 3b) show extended profiles that were independent of the [Au] and an $A_{810}$ that increased with the [Au] (Figure 3c). For instance, $A_{810}$ increased from ≈1.28 to 1.54 when $\phi_{\mathrm{Au}}$ was increased from 5 to 10 wt.%. Here, it is important to note that the photothermal efficiency of any type of NIR photothermal heating depends on the optical absorption in the NIR regime and these results suggest that increasing the [Au] will increase absorbance, leading to high heat generation. Photographs of the photoseeds—AuNP-0 (Figure 3d) and AuNP-10 (Figure 3e)—that was used in our photothermal experiments. FTIR spectroscopy (Figure 3f) revealed the characteristic bands of AuNP-0 at V (7.56–8.64 m${}^{-1}$), W (10.10–10.57 m${}^{-1}$), X & Y (12.57 & 14.11 m${}^{-1}$), and Z (29.50–29.62 m${}^{-1}$) corresponding to -CH${}_{3}$ rocking and Si-C stretching in Si-CH${}_{3}$, Si-O-Si stretching, -CH${}_{3}$ deformation in Si-CH${}_{3}$ and asymmetric -CH${}_{3}$ stretching in Si-CH${}_{3}$, respectively. These bands were not affected by the addition of NPs. This suggests that no primary chemical bonds were formed during the nanocomposite preparation process.

Photothermal heating in aqueous solution. The photothermal heat generation performance of the photoseeds in aqueous solution was evaluated based on initial (after t=30 s) heating rate, d*T*/d*t*, temperature rise, ΔT, and the corresponding η, following 5 min of irradiation at two different $P_{0}:0.5\;\mathrm{and}\;1.0$ W). Table 3 and Figure 4 summarize the results. As expected, d*T*/d*t* and ΔT increased with $P_{0}$ and [Au] (Figure 4a,b). For instance, d*T*/d*t* increased from 0.03 to 0.08 ${}^{\circ }$C/s (ΔT: 8.32 to 17.84 ${}^{\circ }$C) when the $P_{0}$ used to irradiated AuNP-5 was increased from 0.5 to 1.0 W (Table 3). Additionally, the heating rate increased from 0.03 to 0.06 ${}^{\circ }$C/s (ΔT: 8.32 to 15.20 ${}^{\circ }$C) for AuNP-5 and AuNP-10, respectively when they were irradiated at $P_{0}=0.5$ W (Table 3). Furthermore, it can seen from Figure 4c that η increased with $P_{0}$ but decreased with [Au].

Photothermal Heating of Cells Suspension. Based on the results of photothermal heating in aqueous solution, we used a AuNP-10 sample (due to its capability to produce desired temperature levels at low $P_{0}$) to evaluate the ability of a photoseed to kill in-vitro MDA-MB-231 breast cancer cells that are suspended in aqueous solutions during irradiation at two different $P_{0}$-0.5 and 1.0 W. Cells suspensions containing no photoseed were used as the control. Calculated cell viabilities (Figure 5a) showed a clear dependence on temperature levels (Figure 5b). The maximum *T* varied as a function of the presence or absence of AuNP as well as $P_{0}$. For instance, CV reduced from ≈91% (no. of cells: total—113, viable: 59, $\Delta T=42{\;}^{\circ }$C) to ≈48.8% (no. of cells: total—113, viable: 59, $\Delta T=49{\;}^{\circ }$C) when $P_{0}$ used to irradiate the cells suspension in the presence of AuNP-10 was increased from 0.5 to 1.0 W. Without the photoseed, CV remained relatively high, ≈97% (no. of cells: total—113, viable: 59, $\Delta T=42{\;}^{\circ }$C) and ≈94% (no. of cells: total—113, viable: 59, $\Delta T=42{\;}^{\circ }$C) for $P_{0}$ of 0.5 and 1.0 W, respectively. The time dependence of the cell death due to elevated temperature levels has been previously reported [41,42].

### 3.2. Computational Analysis

In order to assess the in-vivo performance of the photoseeds, we conducted a parametric study based on $P_{0}$-(0.5–2.0 W; 0.5 W step). Each simulation was run for 15 min.

A comparison of the three cross-sectional (xy, zy, zx planes) views of the predicted temperature profiles for the case when $P_{0}=1$ W (Figure 6a) showed to be a nonuniform distribution with the maximum occurring around the photoseeds and within the boundary that formed by the simple 2×2 photoseeds configuration (see Figure 6a). It can also be observed that the geometry of the lesion was different for each cross-sectional view. This suggests a dependence of the geometry on the configuration of the photoseeds within the tissue. Furthermore, the temperature at the center of the boundary formed by the photoseeds (P1, cf. Figure 2) rose sharply and then approached a plateau ≈3 min after the laser beam was turned on (Figure 6b). As expected, the heat spread radially outward into the tissue surrounding the photoseeds boundary (Figure 6c). This is consistent with heat transfer by conduction.

It can be seen from the comparison of the cross-sectional (zx plane) views of the predicted thermal dose profiles (Figure 7) for the four different laser powers that the size of the area where complete thermal damage (Ω=1) occurred increased with $P_{0}$. It moved from virtually no damage when $P_{0}=0.5$ W to just around the photoseeds when $P_{0}=1.0$ W to covering the entire area enclosed by the photoseeds and a little bit beyond when $P_{0}=1,5$ W to a perimeter that surrounds the photoseeds with a bigger interval when $P_{0}=2.0$ W than for the case when $P_{0}=2.0$ W. Here, it is important to note that using the laser beam power increases to enlarge the lesion size must be done cautiously to mitigate the deleterious effects of the laser beam on adjacent tissue as shown for the case when $P_{0}=2.0$ W (see Figure 6a). Additionally, it can be observed from Figure 7b,c that the time required to attain Ω=1 decreased and the size of lesion increased, respectively, with increasing $P_{0}$.

Model Validation. The FEM predictions were consistent with the experiments, where the temperature (measured around the photoseed) increased with time. The duration of prediction temperatures were also in good agreement with the experimental results (Figure 8).

## 4. Discussion

Recent efforts to develop mulitfunctional nanocomposite materials have presented engineers with devise opportunities to design novel strategies that are opening up new frontiers in cancer treatment. Furthermore, laser-based technologies provide desirable benefits over other conventional modalities, such as radiation, microwave, radiofrequency, and ultrasound. They offer precise selectivity in heating with more control over blood perfusion and shorter treatment duration. Additionally, they are easier to setup and can be easily coupled with when imaging devices with minimal image artefacts [43]. The combination of these two technologies in the manner present here and previously reported elsewhere can potentially overcome the challenges associated with the thermoseed technique.

N-PTT strategies depend on an interplay between several factors, such as the photothermal (optical) capabilities of the nanocomposites, the configuration of the implants within the target tissue, and the irradiation protocols (e.g., power, shape, duration cross-section, and direction of the laser source) [18]. Here, we performed several experiments to explore key aspects of these factors. We showed the feasibility of fabricating the photoseeds and their ability to generate heat when irradiated with NIR (810 nm) laser in aqueous solutions and in-vitro cell suspensions. Generally, the trend of *T* that is observed in these experiments is in agreement with measured absorbance properties and also consistent with previously reported studies [44]. For the case of the anomalous reciprocal relationship between η and [Au], according to Qin et al. [45] a combination of small size (<30 nm) and low concentration leads to high η. However, increasing [Au] may lead to the formation of clusters, due to the high surface area to volume ratio of the NPs. Therefore, it is reasonable to state here that the η of AuNP-10, which had a higher NP concentration, was lower than AuNP-5, because clusters in AuNP-10 acted as large particles to increase scattering at the expense of absorption.

For such remotely controlled interstitial techniques, patient safety should be taken into consideration during the selection of laser parameters, since high doses can cause collateral damage to healthy tissue [46]. Computational modeling is an essential part of N-PTT strategies, because it enables treatment, wplanninghich is usually aimed at customizing treatment parameters for optimal results [47]. Here, we used the smallest building block (2×2 array) of photoseed configuration in the tissue to show that the predicted lesions spread radially outwards from within the array boundaries, where the highest temperature occurs. Generally, this result is similar to those that were reported for thermoseeds. For instance, Chen et al. [30] performed a 3D computational study of ferromagnetic implant hyperthermia using a fixed 4×4 array and showed that the higher temperature occurred within the region containing the thermoseeds and spread radially outwards. Dughiero et al. [3] found that the geometry of the lession was not affected by configuration when they compared the results of a uniform 3×3 array and a nonuniform arrangement of 20 thermoseeds. It is important to note that the lesion size can be altered by material properties, which, for the case of nanocomposite, can be achieved through the manipulation of the concentration of NPs and the kind of polymer. The direct heating of the tissue by the laser beam, as predicted by the FEM model, can be exploited to reduce the number of photoseeds used.

We acknowledge that the Au-PDMS nanocomposite model system that was used in our study does not have the desired thermal self-regulating feature of the ferromagnetic metal alloys. Au NPs were mainly used because they have been the prime candidates for photothermal therapy [26]; however, several groups have recently shown Fe${}_{3}$O${}_{4}$ NPs to exhibit photothermal properties [18,19]. In addition to their approval for biomedical use by the United States Food and Drug Administration, there is an opportunity to manipulate their composition during synthesis to obtain CTTs that meet the requirements for thermotherapy [48,49]. This offers the opportunity for thermal self-regulating features to be to incorporated in the photoseeds. Furthermore, recent studies that have explored the simultaneous application (DUAL-mode) of both NIR laser and alternating magnetic field (AMF) to the Fe${}_{3}$O${}_{4}$ NPs have shown promising and interesting results. The studies found that the amount of heat that was generated with DUAL-mode equaled the sum of the heating for NIR laser or AMF only [19]. The essence of these results is that the use of the DUAL-mode can potentially lead to a reduction in the number of implants needed to achieve therapeutic temperature levels.

Finally, the results that are presented here provide useful insights that offer a context within which the feasibility of the photoseeds for IT can be discussed. However, we acknowledge that extensive experimental work, including in-vivo studies, is needed to obtain a realistic assessment of the actual performance of this novel approach.

## 5. Conclusions

In summary, we used a combination of experiments and models to demonstrate the feasibility of using photothermally heated photoseeds for interstitial therapy. Experimentally, we found that the photoseeds were capable of generating plasmonic heat under NIR laser irradiation to increase the temperatures in two different media—aqueous solution and cell suspensions. The maximum *T* values obtained were shown to be a function of parameters, such as NP concentration and laser power. Using in-vivo predictions, we showed how the geometry of the lesion is influenced by the laser power and configuration of the implants within the target region. Collectively, the implications of the results are quite significant, since such N-PTT strategies provide several opportunities to overcome the challenges that are associated with conventional thermoseed technique. For instance, photoseed will benefit from a plethora of novel nanomaterials and polymers coupled with facile nanocomposite processing methods and recent advances in laser delivery equipment. Our long term goal is to develop an interstitial thermotherapy technique that is based on photoseeds that will be fabricated at the point of care based on patients data.

## Figures and Tables

**Figure 1 materials-14-00841-f001:**
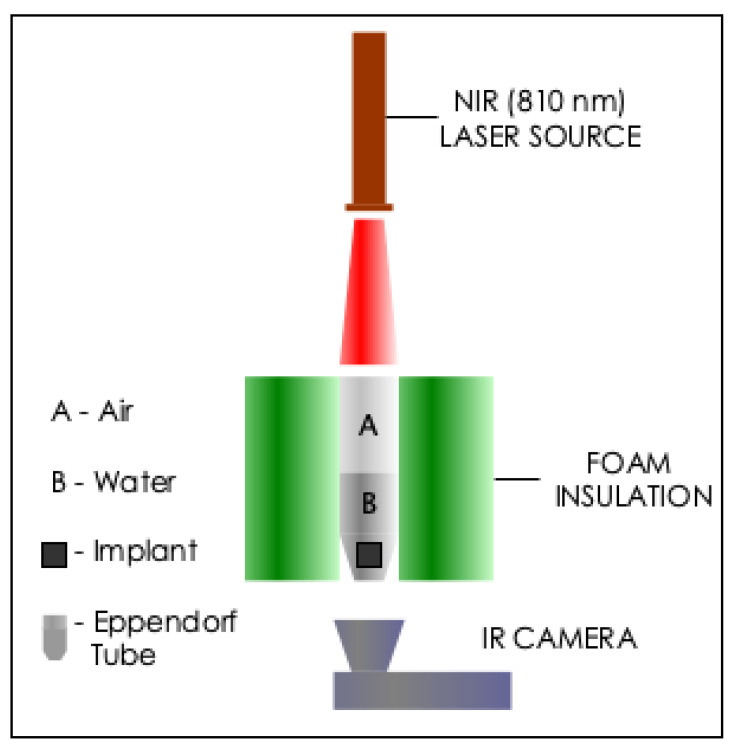
Experimental setup used for the photothermal measurments. The sample (a photoseed and 0.5 mL of deionized water or suspension of cells) contained in a 1.5 mL Eppendorf tube was irradiated by a near-infrared continuous laser at a wavelength of 810 nm.

**Figure 2 materials-14-00841-f002:**
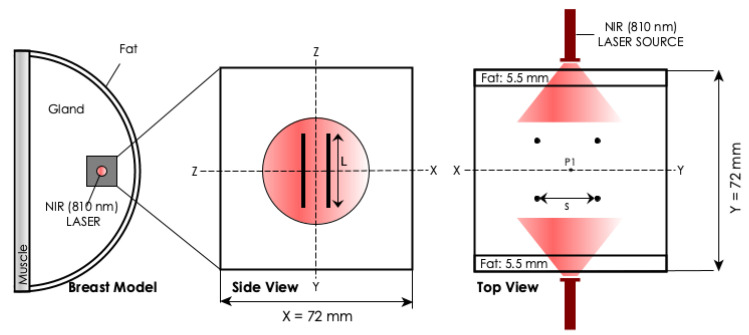
**FEM geometry.** Schematic of the model used to predict the in-vivo photothermal heating. *d* is the distance between adjacent photoseeds and *L* is the length of the photoseed.

**Figure 3 materials-14-00841-f003:**
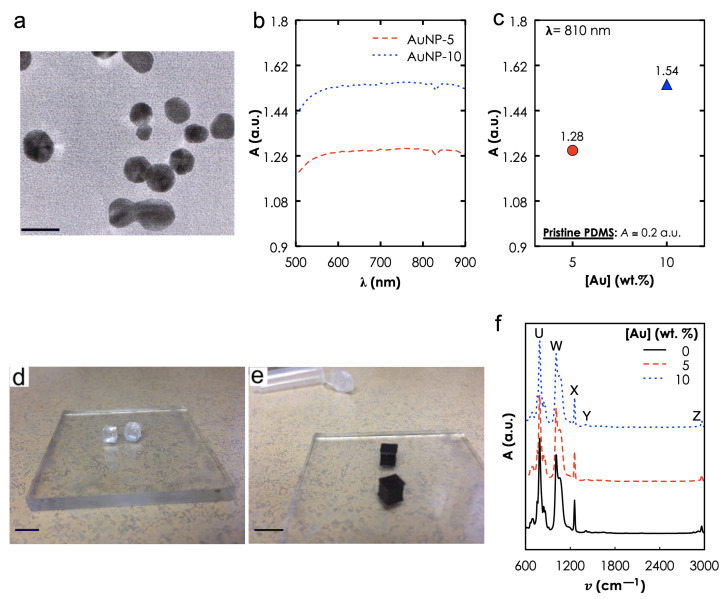
**Materials characterization results**. (**a**) Transmission electron microscopy (TEM) of the Au nanoparticles (AuNPs), Scale bar: 30 nm, (**b**) UV-vis-near infrared (NIR) absorbance spectra of the Gold and Poly dimethylsiloxane (Au-PDMS) nanocomposites at different [Au]: 5 and 10%. Absorbance as a function of (**c**) wavelength (λ) and (**d**) [Au] at λ = 810 nm. Photographs of fabricated samples (Scale bar: 4.5 mm): (**d**) AuNP-0 and (**e**) AuNP-10. (**f**) Fourier transform infra-red of all three samples.

**Figure 4 materials-14-00841-f004:**
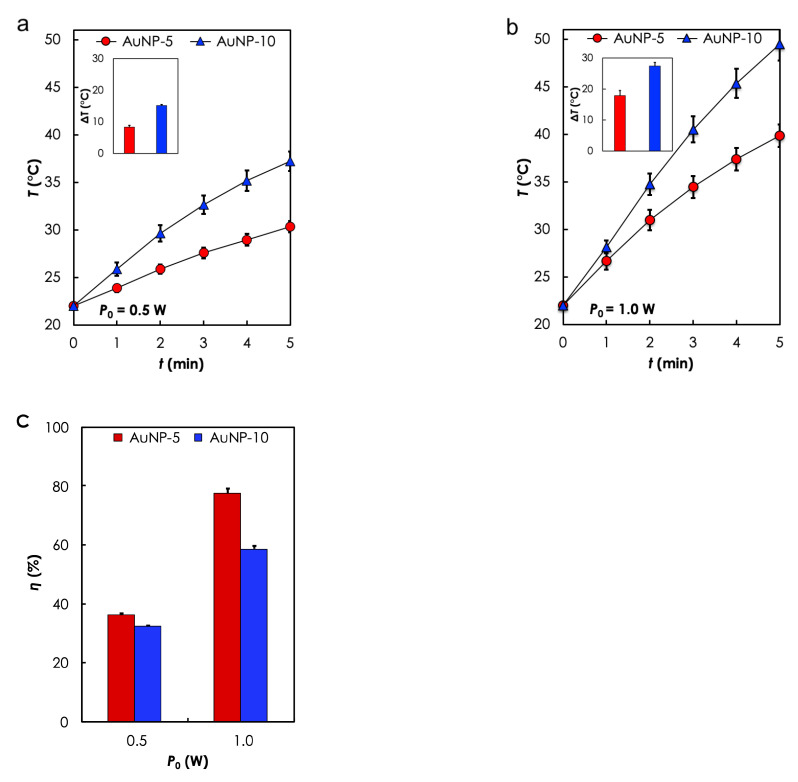
**Photothermal****heating in aqueous solution results**. Temporal response curves for the different samples when (**a**) *P*_0_ = 0.5 W and (**b**) *P*_0_ = 1.0 W. (**c**) Conversion efficiency (η) as a function of *P*_0_ for different *P*_0_ W. Exposure time: *t* = 5 min.

**Figure 5 materials-14-00841-f005:**
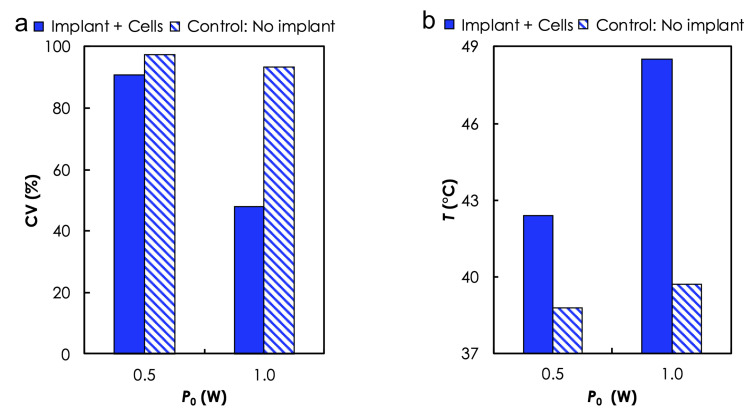
**Photothermal heating in cell suspensions results**. (**a**) In vitro cell viability assessed with Trypan blue dye and (**b**) the corresponding temperature change as a function of laser power after 5 min of irradiation.

**Figure 6 materials-14-00841-f006:**
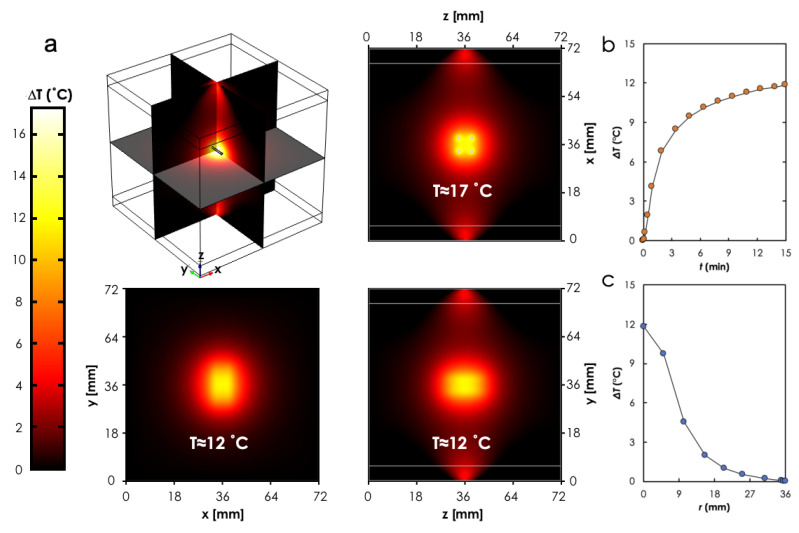
**FEM simulation results.** (**a**) Predicted temperature distributions for different cross-sectional views of the geometry. Temperature change as a function of (**b**) time at and (**c**) distance away from point *P*1. Study parameters: *P*_0_ = 1.0 W, *t* = 15 min.

**Figure 7 materials-14-00841-f007:**
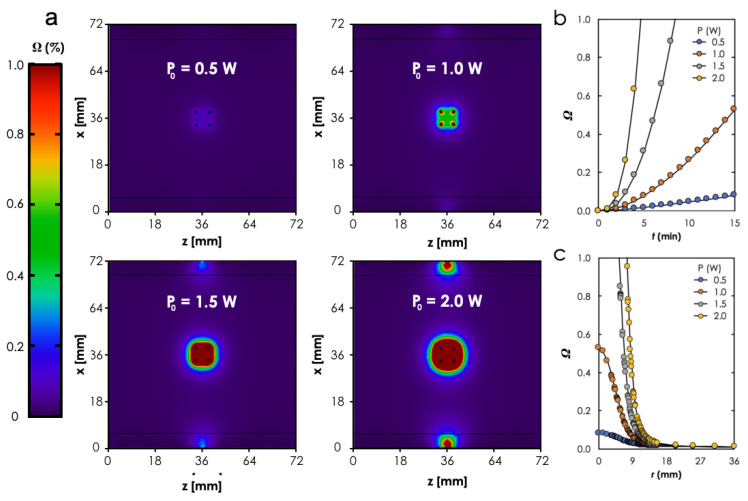
**FEM Simulation results.** (**a**) Comparison of the cross-sectional views (zx-plane) thermal distributions for different *P*_0_. Thermal damage as a function of (**b**) time at and (**c**) distance away from point *P*1. Study parameters: *P*_0_ = 0.5–2.0 W, *t* = 15 min.

**Figure 8 materials-14-00841-f008:**
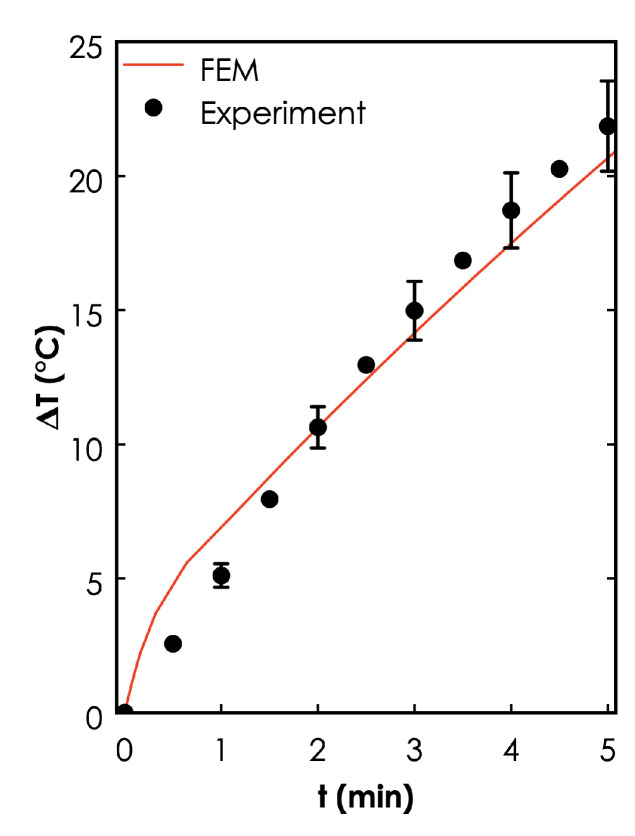
**FEM model validation results**. Comparison of the temporal response curves for the FEM predictions and experimental data for a 5 min irradiation of a cubic photoseed ($\phi_{au}=10\%$ wt.) in 0.5 mL of distilled water contained in a 1.5 mL Eppendorf tube.

**Table 1 materials-14-00841-t001:** The optical properties of the biological domains that were used in the simulations. The values were obtained from Ref. [34].

Tissue	Coefficients, (m${}^{-1}$)	
	Absorption, $\mu_{a}$	Reduced Scattering, $\mu_{s}^{\prime }$
Fat	3	950
Gland	6	1100

**Table 2 materials-14-00841-t002:** Thermo-physical properties of the biological domains that were used in the simulation. The values were obtained from Refs. [38,39].

Tissue	*c* [J (kg K)${}^{-1}$]	λ [W (mK)${}^{-1}$]	ρ [kg m${}^{-3}$]	$Q_{\mathrm{met}}$ [W m${}^{-3}$]	$\omega_{b}$ [s${}^{-1}$]
Fat	2348	0.21	911	400	0.0002
Gland	2960	0.48	1041	700	0.0005
Blood	3617	-	1050	-	-

**Table 3 materials-14-00841-t003:** A summary of the results from photothermal measurements expressed as the mean values of initial (t=30 s) heating rate, dT/d*t*, temperature change, ΔT, calculated as the difference between the initial (t=0 min) and final (t=5 min) temperatures and η, photothermal conversion efficiency, calculated while using Equation (Equation 1).

Sample	[Au] (wt.%)	$P_{0}=0.5$ W				$P_{0}=1.0$ W		
		dT/d*t* (°C/s)	ΔT (°C)	η (%)		dT/d*t* (°C/s)	ΔT (°C)	η (%)
**In Aqueous Solution**									
AuNP-5	5	0.03	8.32	29.61		0.08	17.84	63.47
AuNP-10	10	0.06	15.20	26.38		0.09	27.46	47.66
No photoseed	-	-	1.30	-		-	2.70	-
**In-vitro Suspension**									
AuNP-10 + Cells	10	-	5.40	-		-	11.5	-
Cells only	-	-	1.80	-		-	2.70	-

## Data Availability

The data presented in this study are available in article.

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
