# Peer review of "Plasmonic Nanocomposite Implants for Interstitial Thermotherapy: Experimental and Computational Analysis"

_materials, 2021, doi:10.3390/ma14040841_

Round 1
Reviewer 1 Report
Reviewer comment
Manuscript ID: materials-1087396
Title: Plasmonic Nanocomposite Implants for Interstitial Thermotherapy: Experimental and Computational Analysis.
The authors reported the Plasmonic Nanocomposite Implants for Interstitial Thermotherapy: Experimental and Computational Analysis. After careful reading, I feel that the work is not acceptable for publication in materials in its present form. The following issues should be addressed.
Comments
- Provide more detail for introduction part in this work, it is useful for readers.
- The experimental section should be detailed especially for the model validation and implementation.
- Provide more detail for discussion part.
- Please provides the references for all equations and formula.
- Fig. 3a, is not clear make clear.
- Conclusions: the authors need to improve with more specific short results and conclusions, i.e. academic novelty or technical advantages.
- References: author should use order and there are recent references in 2019 and 2020 treating the same subject, you can use. Make all references in same format for volume number, page.
- Furthermore, they should add the graphical abstract, it is use full to readers.
- Several faults: are added or missing spaces between words: see manuscript file.
- Some sentences need reconstruction and the level of English should be improved.
Reviewer 2 Report
Konku-Asase and Kan-Dapaah present an experimental and computational analysis on the use of plasmonic nanocomposites based on Au nanoparticles and PDMS for their use as implants for interstitial thermotherapy. The work is well written and presented, and it may be of interest to a broad audience. However, I found some inconsistencies while reading and it is for this reason that I cannot recommend this manuscript for publication in its current form.
The introduction gives enough information to understand the context and the research questions to be addressed, but it fails to highlight most recent research advances on the topic and the limitations, therefore the motivation for this work is weaken. I am referring in particular to references [3] to [13]. There are no references either to the use of Au/PDMS nanocomposites for this or other applications, and that’d be appreciated. Also, the sentence in line 59 is not clear to me, what do the authors mean?
Figure 1 is not referred to in the text, so it’s difficult to understand. A more detailed caption would help too.
In the experimental section, there are some details missing for other researchers to reproduce the study. It is not clear what the sizes of cubic acrylic moulds are, and this is important to understand later the absorbance measurements and the in vivo experiments. It is unclear how the composites are prepared, and how the 5% or 10% concentration of NPs is calculated. I am assuming the AuNPs were in a colloidal dispersion? It would also help to know how the gold nanoparticles were prepared or the specifications in the product if it was purchased. What is the reason behind using 5% or 10% NP concentration? In Line 95 it is stated that the laser spot diameter was 2cm, which is much larger than the 1.5 mL Eppendorf tube… This does not make sense.
In the results section, the authors claim the AuNPs have 28 nm diameter and they refer to Figure 3a. This figure does not show such average diameter, which seems to clearly be below 20 nm. The absorbance measurements are also controversial. First of all, they are above 1 which may mean lack for accuracy, but also it is not clear how this is experimentally done. The absorbance curve for only AuNPs in solution would be appreciated too. The discussion around the increase in absorbance for increasing the heat is weak, why wouldn’t you increase the concentration of AuNPs then? The conclusions extracted from the FTIR graph are also poor. I can only see there PDMS, therefore the 3 graphs are redundant and do not provide any information with regards to the addition of AuNPs. Line 172, that may be redundant, what bonds would be formed?
In Table 3 and subsequent paragraphs, dT/dt is measured in °C. This is wrong, as it is my understanding that this is a measurement of the variation in temperature with time, either minutes or seconds.
With regards to the in vitro experiments, I am missing controls with healthy cells. How does the photothermal heating affect non-cancerous cells?
Round 2
Reviewer 1 Report
Reviewers' comments:
The manuscript can published. The authors have answered the questions. So that I recommended this manuscript accept for publication in Materials.
Author Response
Thank you
Reviewer 2 Report
The authors have addressed correctly most of my comments, and I would recommend the paper for publications after addressing minor comments:
When I was asking about other works that combined Au and PDMS I was enquiring about other works that have reported the use of this hybrid for thermotherapy, and if it hasn't been used, what are the reasons behind (taking into account the widespread use of both PDMS and AuNPs). Any comment about the biocompatibility of PDMS?
Line 69: typo. Modify "to improved" by "to improve".
Caption figure 1: "at a wavelength": at what wavelength?
Author Response
When I was asking about other works that combined Au and PDMS I was enquiring about other works that have reported the use of this hybrid for thermotherapy, and if it hasn't been used, what are the reasons behind (taking into account the widespread use of both PDMS and AuNPs).
To the best of our knowledge, there aren't any reported studies that have used the Au-PDMS nanocomposites, at least in the manner we proposed, for thermotherapy. So we have provided separate comments in the last paragraph of the Introduction section to support our decision to use them to prepare the nanocomposites. For the case of the Au NPs, we included why they are the primary candidate for PTT and provided references to the plethora of fundamental research and recently pilot clinical studies reported in the literature.
Any comment about the biocompatibility of PDMS?
For the case of the PDMS, we have included comments on their attractive properties to support the reasons for their widespread use in several biomedical applications.
Line 69: typo. Modify "to improved" by "to improve".
Thank you for the suggestion. We have made the correction accordingly.
Caption figure 1: "at a wavelength": at what wavelength?
Thank you for the suggestion. We have now added the value of wavelength.